# Point-of-Care Ultrasound Within One Hour Associated with ED Flow and Resource Use in Non-Traumatic Abdominal Pain: A Retrospective Observational Study

**DOI:** 10.3390/diagnostics15131580

**Published:** 2025-06-21

**Authors:** Sheng-Yao Hung, Fen-Wei Huang, Wan-Ching Lien, Te-Fa Chiu, Tse-Chyuan Wong, Wei-Jun Lin, Shih-Hao Wu

**Affiliations:** 1Department of Emergency Medicine, China Medical University Hospital, Taichung 404327, Taiwan; 2School of Medicine, College of Medicine, China Medical University, Taichung 404328, Taiwan; 3Department of Emergency Medicine, National Taiwan University Hospital, Taipei 100008, Taiwan; 4College of Medicine, National Taiwan University, Taipei 100233, Taiwan; 5College of Public Health, China Medical University, Taichung 406040, Taiwan

**Keywords:** ultrasonography, clinical approach, mortality, intensive care unit, return visit, computed tomography, decision-making, length of stay, costs, ED crowding

## Abstract

**Background:** Although the value of point-of-care ultrasound (PoCUS) is well-established for specific diseases and in the hands of trained users, its broader impact on overall ED efficiency is not yet fully known. This study aims to evaluate the association of early PoCUS, performed within 1 h of presentation, with ED patient flow, healthcare resource utilization, and quality of care in adults with non-traumatic abdominal pain. **Method:** This retrospective cohort study included 44,863 adult patients (≥18 years) presenting with non-traumatic abdominal pain from January 2021 to December 2023. Patients were grouped into PoCUS and no-PoCUS categories, with a subgroup analysis for those receiving PoCUS within 1 h, to evaluate ED LOS, and costs for different ED dispositions. Outcomes measured included hospital LOS, costs, mortality, and ICU admission. **Results:** The mean age of the subjects was 44.4 ± 17.9 years, and 61.2% were female. PoCUS was performed in 39.7% of cases, with 69.6% of these conducted within one hour. Additionally, 30.5% underwent CT. The PoCUS group had a significantly shorter ED LOS compared to the no-PoCUS group among patients admitted to general wards (*p* < 0.001), but not in outpatient dispositions (*p* = 0.282) or ICU admissions (*p* = 0.081). Subgroup analysis of patients receiving PoCUS within 1 h showed a significantly shorter LOS for both outpatient dispositions (*p* < 0.001) and general ward admissions (*p* < 0.001), with no effect on ICU admissions (*p* = 0.869). The presence or absence of CT did not alter these findings. Multivariable analysis indicated that patients who received PoCUS within one hour alone at index visit and admitted after an unscheduled return visit had lower initial ED costs (−9436.1 TWD, *p* < 0.001) and shorter ED LOS (−11.59 min, *p* < 0.001) than patients admitted directly at the index visit, with no significant increase in total resource utilization or adverse outcomes after return visits. **Conclusions:** PoCUS, especially when performed within one hour, was associated with reduced ED LOS and healthcare resource utilization for both outpatient dispositions and inpatient admissions without compromising patient safety or quality of care.

## 1. Introduction

Non-traumatic abdominal pain is a common and often complex presentation in the emergency department (ED), accounting for nearly 7% of all ED visits in the United States and representing more than 3 million patient encounters [1]. It encompasses a wide range of potential causes, from benign, self-limiting conditions to life-threatening emergencies such as bowel obstruction, mesenteric ischemia, or ruptured abdominal aortic aneurysm [2]. The diagnostic process for these patients is often challenging and time-consuming [3,4], which can lead to increased length of stay (LOS) and higher healthcare resource utilization. Traditionally, ED diagnostic workups for abdominal pain rely on clinical assessment, laboratory tests, and imaging studies [5]. While computed tomography (CT) remains a widely used tool due to its high diagnostic accuracy [6], its use in the ED can result in significant delays [7], increased radiation exposure [8], and higher costs. As the demand for emergency care continues to rise, optimizing patient flow and minimizing healthcare costs have become critical goals for improving ED efficiency and patient care.

Point-of-care ultrasound (PoCUS) has emerged as a valuable diagnostic tool in emergency medicine, offering rapid, non-invasive, real-time imaging directly at the bedside [9,10]. PoCUS is a bedside diagnostic tool that allows clinicians to acquire, interpret, and immediately integrate ultrasound imaging into patient care [10]. PoCUS has been successfully used to diagnose a variety of abdominal conditions, such as gallbladder disease [11], kidney stones [12], and ectopic pregnancy [13,14], with high sensitivity and specificity. Furthermore, previous studies of the impact of PoCUS on length of stay (LOS) and costs have focused on specific diseases or complaints [15,16,17]. However, few have examined its effects on all patients presenting to the ED with non-traumatic abdominal pain as their main symptom. PoCUS enables immediate diagnostic assessments, facilitating timely clinical decision-making [15,17]. Nevertheless, the association between PoCUS execution time and patient flow, as well as the utilization of medical resources for different ED dispositions, remains unclear.

The benefits of PoCUS must be weighed against other imaging modalities. Computed tomography (CT) provides superior diagnostic accuracy [18] and has been shown to enhance surgical decision-making in cases of acute abdominal pain [19]. However, CT carries notable drawbacks, including higher costs and significant radiation exposure [17]. In contrast, PoCUS is more accessible and avoids radiation but is highly operator-dependent [20]. Its diagnostic performance varies with patient factors, operator expertise, and equipment quality [21].

The timing of PoCUS administration may play a crucial role in improving ED efficiency. Early PoCUS, particularly when performed within the first hour of patient presentation, was associated with shorter ED LOS and earlier surgical consultation, enhancing ED efficiency in patients with mild acute cholecystitis [22]. However, it remains unclear whether PoCUS can streamline ED management processes, such as reducing ED LOS and healthcare resource utilization, particularly in the context of non-traumatic abdominal pain, thereby potentially mitigating ED crowding.

This study aims to evaluate the association of early PoCUS, performed within 1 h of presentation, with ED patient flow, healthcare resource utilization, and quality of care in adults with non-traumatic abdominal pain.

## 2. Methods

### 2.1. Study Design

This retrospective cohort study used data from the electronic medical record (EMR) of a tertiary medical center in Taiwan with more than 160,000 ED visits annually, between 1 January 2021 and 31 December 2023. The database is de-identified but contains a unique, encrypted personal identifier that allows researchers to link claims between ED and inpatient databases. This study was approved by the Institutional Review Board of China Medical University Hospital (CMUH113-REC2-008), and informed consent was waived. STROBE guidelines for observational studies were followed, and all elements in the checklist for cross-sectional studies are presented in content and structure [23].

### 2.2. Setting and Population

The patients were included if they (i) had an ED visit with no prior ED visit or hospitalization in the preceding three days; (ii) had non-traumatic abdominal pain as the chief complaint at triage; (iii) were adults aged 18 years or older. Patients were excluded if they were (i) younger than 18 years; (ii) had traumatic abdominal pain; (iii) were transferred from other EDs; or (iv) were discharged against medical advice from other clinics or hospitals. This study included 44,863 patients.

This tertiary center employed 42 emergency medicine specialists and 19 residents, with all attending physicians certified in emergency medicine. Since 2012, PoCUS has been a core component of ultrasound training in emergency medicine residency programs in Taiwan, with all residents undergoing hands-on assessments led by certified instructors from the Taiwan Society of Ultrasound in Medicine, each with over a decade of experience. In clinical settings, PoCUS was performed by residents or attending emergency physicians, all of whom regularly complete refresher courses to maintain and advance their skills.

The ED is equipped with one mobile GE Vivid iq, three mobile GE Venue units, three mobile GE LOGIQ™ e units, one fixed GE LOGIQ E9 XDclear 2.0 system, and one fixed GE LOGIQ S7 General Imaging system. All machines undergo routine maintenance every three months, and technical support is available immediately in case of malfunction. Ultrasound images can be uploaded wirelessly to the Picture Archiving and Communication System (PACS) in real time. Reports can be typed directly into the Hospital Information System (HIS), ensuring efficient documentation and accessibility.

In this study, PoCUS was not limited to evaluating a specific organ or confirming a diagnosis. Instead, it was used to address the clinical questions that arise during patient care, often requiring a comprehensive assessment across multiple organs and systems based on the patient’s presenting symptoms. All physicians at this tertiary center had received standardized PoCUS training and could apply any protocol or technique they had learned to guide their clinical decisions. We studied patients who received PoCUS versus those who did not and analyzed data on the number of unscheduled return visits with admission in each group to compare ED LOS and costs.

An index ED visit was defined as one with no prior ED visit or hospitalization in the preceding three days. A return visit referred to any ED revisit occurring within 72 h of discharge from the index visit; for patients with multiple revisits in that period, only the first was included. The unit of analysis was the visit, allowing individual patients to contribute multiple index visits during the study period. We focused on early rather than late revisits, as these are generally more preventable and more responsive to hospital-based interventions [24]. The cohort was divided into two groups for comparison according to the presence or absence of PoCUS at the index visit. The PoCUS group was further divided into (1) PoCUS performed within 1 h and (2) PoCUS performed between 1 and 2 h after the ED visit, to investigate the time effect of PoCUS on ED resources and patient flow.

### 2.3. Variables

The EMR contains information on patient demographics, visit date and time, triage level, comorbidities, PoCUS, time of PoCUS, CT, ED disposition, ED length of stay (LOS), ED costs, admission costs, and hospital LOS. The Taiwan Triage and Acuity Scales system is a computerized, five-level system with acuity levels 1 to 5 indicating resuscitation, emergent, urgent, less urgent, and non-urgent, respectively [25].

### 2.4. Outcome Measures

The outcome measures were ED LOS, ED costs, inpatient mortality, intensive care unit (ICU) admission, hospital LOS, and total ED and inpatient costs in TWD (New Taiwan dollar).

The ED LOS was defined as the period from the patient’s initial presentation to the ED, as documented by the triage nurse, to the patient’s discharge from the ED. ED LOS was calculated as the following five points: discharge from the ED, discharge from the observation room, admission to the general ward, admission to the ICU, and ED mortality. The hospital LOS of patients admitted to the general ward or ICU was documented as a secondary outcome to evaluate the prognosis of patients.

### 2.5. Data Analyses

Summary statistics are presented as means (with standard deviations). We examined bivariate associations using Student’s t-test and chi-square tests, as appropriate. The clinical outcomes (mortality and ICU admission) and resource use (LOS and cost) were analyzed by comparing the PoCUS group to the NO-PoCUS group, and the PoCUS within 1 h group to the NO-PoCUS group with and without CT, respectively. Missing data were limited to vital signs and anthropometric measures (i.e., weight, height, and BMI), which were excluded from calculations of means and SDs.

We used multivariable logistic and linear regression models to adjust for differences in patient mix. Potential confounding factors included age, gender, triage, BMI, and comorbidities. All odds ratios (ORs) and beta-coefficients are presented with 95% CIs. We performed all analyses using SAS software (version 9.4; SAS Institute Inc., Cary, NC, USA). All *p* values are two-sided, with *p* < 0.05 considered statistically significant.

## 3. Results

### 3.1. Population Distribution

A total of 44,863 index ED visits were entered in the core analyses (Figure 1). Among them, 17,819 (39.7%) patients received PoCUS, and 27,044 (60.3%) patients did not receive PoCUS. A total of 12,399 (69.6%) patients received PoCUS within one hour, and 3102 (17.4%) received PoCUS between one and two hours. A total of 13,670 (30.5%) patients received CT, and 31,193 (69.5%) patients did not receive CT.

### 3.2. Baseline Demographics

Table 1 provides the baseline demographics of patients who received PoCUS compared to those who did not. Compared with the No-PoCUS group, patients who received PoCUS were significantly younger, predominantly female, and more likely to be triaged at level 3. In terms of medical resource utilization characteristics, patients who received PoCUS had shorter ED and hospital LOS, fewer CTs performed, and fewer admissions, ICU admissions, and deceased patients. In addition, patients who received PoCUS were more frequently discharged with outpatient disposition (OPD). These results suggest that PoCUS is more likely to be used in lower disease severity.

### 3.3. Impact of PoCUS Timing on ED LOS and Costs

Table 2 illustrates the impact of PoCUS execution time on ED LOS and costs across different dispositions. Among the entire population and the populations with or without CT, the PoCUS group demonstrated significantly lower ED LOS compared to the No-PoCUS group, only in patients admitted to the ordinary ward. However, the PoCUS performed within the 1 h group demonstrated significantly lower ED LOS compared to the No-PoCUS group, both in patients discharged with OPD and admitted to the ordinary ward. In all patients admitted to the ICU and in the group with PoCUS performed between 1 and 2 h, there was no difference in ED LOS. These revealed that the impact of PoCUS on reducing ED LOS would be enhanced by PoCUS performed within 1 h. Nevertheless, the effect disappeared in patients admitted to the ICU.

Among the entire population and the population without CT, ED costs were significantly higher in the PoCUS group for patients discharged with OPD, regardless of when PoCUS was performed. However, among the population without CT, ED costs were significantly lower in the PoCUS group for patients admitted to the ordinary ward and in the PoCUS group performed within the 1 h group for both patients admitted to the ordinary ward and the ICU. Among the population with CT, ED costs were not different for patients admitted to the regular ward or ICU. These results indicate that PoCUS would increase ED costs in less severe diseases, whereas PoCUS would reduce ED costs in diseases requiring admission, and the effect could be enhanced by PoCUS performed within 1 h. However, the advantage disappeared with CT.

### 3.4. Quality of Care with POCUS

Table 3 compares quality-of-care metrics between patients admitted after their index visit (*N* = 7967) and those who received PoCUS within one hour alone at the index visit and admitted after an unscheduled return visit (*N* = 110). Except for lower first ED LOS and costs, there were also lower admission costs, total costs, hospital LOS, and ICU LOS in patients who received PoCUS within one hour alone at the index visit and admitted after an unscheduled return visit. However, there was no difference in ICU rate and mortality.

Table 4 outlines the quality of care with PoCUS after adjusting for age, gender, triage, BMI, and comorbidities. Patients who received PoCUS within one hour alone at index visit and admitted after an unscheduled return visit still had a lower first ED LOS (adjusted difference, −11.59, 95% CI, −14.20 to −8.98, *p* < 0.001), and lower first ED costs (adjusted difference, −9436.1 TWD, 95% CI, −11,542.9 to −7329.3, *p* < 0.001); however, no difference in total costs, hospital LOS, ICU admission rate, and mortality. This means that the PoCUS within one hour used fewer medical resources for the index visit without increasing the admission costs or the hospital LOS after an unscheduled return visit, representing no adverse impact on patient safety or quality of care.

## 4. Discussion

This study demonstrated that the association between PoCUS and reduced ED length of stay (LOS) was more pronounced when PoCUS was performed within the first hour. However, this effect was not observed in patients admitted to the ICU. The findings also suggested that PoCUS was associated with lower ED costs in conditions requiring admission, with the greatest reduction seen when performed within the first hour; however, this advantage was no longer evident when CT was used. Patients who underwent PoCUS within the first hour (without CT) during their initial visit and were later admitted after an unscheduled return visit had lower initial ED costs and shorter ED LOS, without a significant increase in overall resource utilization or adverse outcomes upon return. These results are consistent with previous research [26,27], reinforcing PoCUS as a valuable risk stratification tool in emergency medicine for expediting decision-making and optimizing resource allocation.

### 4.1. PoCUS Utilization Patterns

PoCUS was utilized in approximately 40% of ED visits in this study, with higher usage among younger patients, females, and those triaged at level three. This pattern suggests that PoCUS was primarily employed for non-traumatic abdominal pain of moderate severity rather than for critically ill cases, where clinicians more commonly relied on laboratory tests and CT scans for definitive diagnosis. However, PoCUS has several important applications in critically ill patients, including the assessment of abdominal organ function, differentiation of shock states, and identification of septic sources [28]. The higher utilization among younger patients may reflect the relative simplicity of abdominal pain diagnoses in this population, making PoCUS a more reliable aid in clinical decision-making while minimizing CT radiation exposure. A stepwise approach beginning with PoCUS before proceeding to CT is recommended [29]. Additionally, multiorgan PoCUS has demonstrated utility in the assessment and management of geriatric patients [30]. In women with acute pelvic pain, PoCUS serves as the primary imaging modality, offering high accuracy in detecting or ruling out urgent gynecologic conditions requiring immediate surgical intervention [31], which may explain its predominant use in female patients.

PoCUS use was also associated with improved patient flow in lower-acuity cases, including higher rates of outpatient disposition, lower ICU admission rates, and reduced inpatient mortality. These findings support its role as an effective screening tool for managing less severe presentations, potentially reducing the need for CT scans and associated radiation exposure. However, barriers to broader adoption persist, including a lack of remuneration, a high workload, and skepticism among some general practitioners regarding its diagnostic accuracy and clinical value [32,33]. Despite these challenges, studies suggest that physicians who integrate PoCUS into their practice are often more proactive, using it to obtain immediate diagnostic insights and expedite clinical decision-making, contributing to shorter ED length of stay [34]. Enhancing PoCUS education, particularly for its use in critically ill patients with abdominal pain and in the geriatric population, may further support its integration into routine ED practice.

### 4.2. The Association of PoCUS with ED Patient Flow and Resource Utilization

This study found a significant association between PoCUS use and reduced ED LOS, particularly when performed within the first hour of patient presentation. The most notable reductions were observed in outpatient dispositions and those admitted to the general ward. Similar findings have been reported in previous studies, where PoCUS was associated with shorter ED LOS in specific conditions such as small bowel obstruction [17], early pregnancy [35], diverticulitis [16], and acute cholecystitis [22]. However, no significant difference in ED LOS was observed among patients admitted to the ICU. These cases often involve greater complexity, requiring additional imaging, laboratory tests, and interventions, and waiting for consultation and ICU bed availability. While PoCUS remains an important tool in the management of critically ill patients, its impact varies depending on the clinical context. One study reported that POCUS was considered to have the potential to reduce or prevent mortality and morbidity in 45% of cases where it was not used, particularly in cardiopulmonary assessments [36]. However, its effectiveness is closely related to the operator’s experience and skill level, with inadequate training increasing the risk of misdiagnoses, highlighting the need for structured education and competency assessment [37]. In summary, PoCUS demonstrates substantial utility in evaluating abdominal pain in the ED and is associated with improvements in patient flow. Its diagnostic value is influenced by operator expertise and patient characteristics, underscoring the importance of appropriate training and integration into a comprehensive clinical assessment to optimize ED resource utilization.

### 4.3. Quality of Care and Patient Safety

Errors in diagnosis, prognosis, treatment, follow-up care, and information provision often contribute to unscheduled ED returns. Diagnostic errors, in particular, show a strong correlation with adverse outcomes post-discharge, including elevated mortality rates and ICU admissions [38,39,40]. While the reliability of traditional metrics like 72 h returns as indicators of care quality or safety has been debated, a more specific measure—such as 72 h returns requiring admission—demonstrates a clearer link to patient outcomes [41,42,43]. Additionally, shorter ED LOS [44] and insufficient evaluation and treatment [45] at the index visit have been suggested to be risk factors for increased unscheduled return visits. In this study, patients with the shortest ED LOS and lowest costs (less assessment and treatment) at the index visit in the PoCUS within 1 h (without CT) group who were admitted after an unscheduled return visit still did not have significant difference in hospital costs, LOS (total and ICU), and ICU admission or mortality after the return visit compared with direct admission at the index visit. These findings suggest a strong association between PoCUS use and effective risk stratification, supporting safe initial discharge decisions. Even in cases of return visits requiring admission, the quality of care remained consistent, aligning with the safety of PoCUS-guided decision-making.

These findings highlight PoCUS as an excellent risk-stratification tool that supports safe initial discharge decisions. Even when patients returned and required admission, their quality of care remained uncompromised, reinforcing the safety of PoCUS-guided decision-making.

Concerns that PoCUS might be linked to missed diagnoses or delayed interventions [46] are not supported by this study. The admission after an unscheduled return visit may be related to disease progression [47]. These findings highlight PoCUS as a valuable component of risk stratification, enhancing decision-making without increasing patient risk.

### 4.4. Strengths and Limitations

This study analyzes the association between PoCUS performed within the first hour and patient flow, resource utilization, and the quality of PoCUS and care at the index ED visit. Quality is indirectly assessed by evaluating the days and costs of admissions after an unscheduled return visit. In addition, a large sample size and a long period were utilized to ensure reliability. Despite these strengths, this study has several limitations. First, as a retrospective cohort study, it is susceptible to selection bias and potential unmeasured confounders. While adjustments were made for various factors, residual confounding may still influence the observed associations. Second, the study was conducted at a single tertiary medical center in Taiwan, which may limit the generalizability of the results to other healthcare settings with varying PoCUS training, resources, and patient populations. Third, physician-specific factors, such as variations in PoCUS expertise and interpretation, were not accounted for and could influence outcomes. Furthermore, the effect of different training levels on diagnostic accuracy has not been evaluated. However, PoCUS examinations performed by residents were supervised by attending physicians, which may have reduced the impact of differences in training on the results. Fourth, this study did not focus on a specific organ or system, nor did it aim to confirm a diagnosis. Therefore, the specificity and sensitivity of PoCUS for making specific diagnoses were not investigated. Fifth, part of this study was conducted during the peak of the COVID-19 pandemic, when ED visits declined significantly and infection control measures, such as universal precautions and enhanced disinfection, disrupted normal workflows. As a result, the timing of PoCUS may have deviated slightly from standard practice. However, the clinical approach to non-traumatic abdominal pain remained unchanged. Notably, 17,819 patients underwent PoCUS examination, of whom only 34 had a confirmed SARS-CoV-2 infection. No complications were observed in SARS-CoV-2-positive patients who underwent PoCUS. Finally, while the study did not assess PoCUS indications or accuracy, its goal was to evaluate PoCUSwith outcomes in non-traumatic abdominal pain without restricting indications. Outcomes such as hospital LOS, admission costs, ICU admission rates, and mortality post-return visit were used as indirect measures of decision-making qualityto PoCUS use.

## 5. Conclusions

This study provided robust evidence that integrating PoCUS in the ED, particularly within the first hour, was associated with reduced ED LOS and lower admission rates, without compromising the quality of care or patient safety. These findings highlight the role of PoCUS as an effective risk stratification tool for patients with non-traumatic abdominal pain, facilitating clinical decision-making, supporting safe initial discharge, and optimizing patient flow and resource allocation in emergency care. Future research should validate these results across diverse healthcare settings and examine its effectiveness in various diagnostic groups within the ED.

## Figures and Tables

**Figure 1 diagnostics-15-01580-f001:**
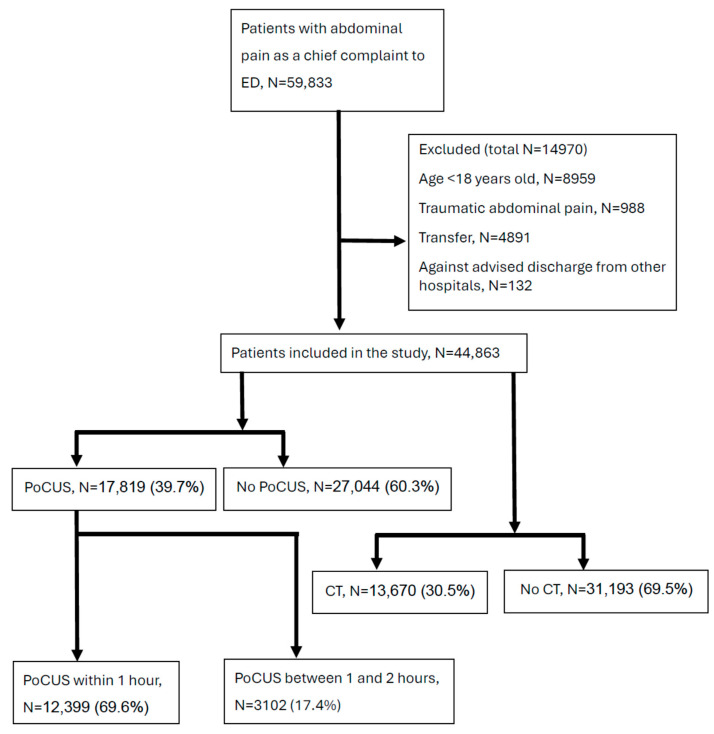
Flow diagram of study population selection.

**Table 1 diagnostics-15-01580-t001:** Demographics of the study population.

	PoCUS, *N* = 17,819	No PoCUS, *N* = 27,044	*p*
Age	42.91 ± 17.80	45.42 ± 19.56	<0.001
Sex			<0.001
Male	6497 (36.46)	10,905 (40.32)	
Female	11,322 (63.54)	16,139 (59.68)	
BMI	24.11 ± 24.98	24.16 ± 37.70	0.855
Triage			<0.001
1	104 (0.58)	340 (1.26)	
2	1684 (9.45)	3555 (13.15)	
3	15,750 (88.39)	22,464 (83.06)	
4	279 (1.57)	634 (2.34)	
5	2 (0.01)	51 (0.19)	
Heart rate	87.43 ± 17.29	91.55 ± 18.37	
Systolic blood pressure	130.2 ± 23.93	128.3 ± 23.46	
Diastolic blood pressure	80.67 ± 14.61	79.39 ± 14.43	
Body temperature	36.64 ± 0.68	36.73 ± 0.88	
Respiratory rate	19.47 ± 1.56	19.58 ± 1.60	
CT	5086 (28.54)	8584 (31.74)	<0.001
Discharged with OPD	13,791 (77.39)	18,456 (68.24)	<0.001
Admission to ward	2406 (13.50)	5324 (19.69)	<0.001
Admission to ICU	69 (0.39)	168 (0.62)	<0.001
Expire in ED	1 (0.01)	20 (0.07)	0.001
ED LOS	5.11 ± 7.12	6.24 ± 9.52	<0.001
Hospital LOS	7.55 ± 10.90	9.40 ± 12.88	<0.001

PoCUS: point-of-care ultrasound; BMI: body mass index; CT: computer tomography; OPD: outpatient disposition; ICU: intensive care unit; ED: emergency department; LOS: length of stay.

**Table 2 diagnostics-15-01580-t002:** The association between PoCUS execution time and ED LOS and costs across different patient dispositions.

Whole Population, *N* = 44,863									
	No PoCUS	PoCUS	*p*	No PoCUS	PoCUS Within 1 h	*p*	No PoCUS	PoCUS Between 1 and 2 h	*p*
OPD, *N* = 32,247	18,456	13,791		18,456	9743		18,456	2403	
LOS in ED (h)	3.80 ± 5.77	3.74 ± 4.60	0.282	3.80 ± 5.77	3.33 ± 3.92	<0.001	3.80 ± 5.77	3.82 ± 4.38	0.849
Costs in ED (NT$)	4367.9 ± 4216.2	4918.9 ± 3931.7	<0.001	4367.9 ± 4216.2	4657.2 ± 3738.8	<0.001	4367.9 ± 4216.2	5157.2 ± 4240.9	<0.001
Admission to ward (no ICU), *N* = 7730	5324	2406		5324	1542		5324	437	
LOS in ED (h)	16.13 ± 14.56	14.78 ± 13.03	<0.001	16.13 ± 14.56	13.28 ± 11.76	<0.001	16.13 ± 14.56	16.26 ± 13.75	0.857
Costs in ED (NT$)	13,221.1 ± 9812.8	13,064.6 ± 9515.9	0.512	1322.1 ± 9812.8	12,820.6 ± 9169.8	0.137	13,221.1 ± 9812.8	13,571.4 ± 10,002.0	0.473
Admission to ICU, *N* = 237	168	69		168	43		168	15	
LOS in ED (h)	12.94 ± 13.26	16.40 ± 16.01	0.087	12.94 ± 13.26	13.32 ± 14.03	0.869	12.94 ± 13.26	15.59 ± 15.64	0.466
Costs in ED (NT$)	33,873.2 ± 36,233.5	33,246.1 ± 30,747.5	0.899	33,873.2 ± 36,233.5	26,144.8 ± 19,990.3	0.064	33,873.2 ± 36,233.5	32,197.0 ± 30,001.0	0.862
Without CT, *N* = 31,193									
	No POCUS	POCUS	*p*	No POCUS	POCUS within 1 h	*p*	No POCUS	POCUS above 1 h	*p*
OPD, *N* = 26,008	14,857	11,151		14,857	7865		14,857	1987	
LOS in ED (h)	3.16 ± 5.27	3.12 ± 4.09	0.577	3.16 ± 5.27	2.79 ± 3.43	<0.001	3.16 ± 5.27	3.14 ± 3.65	0.901
Costs in ED (NT$)	2906.6 ± 2864.2	3576.8 ± 2675.2	<0.001	2906.6 ± 2864.2	3320.9 ± 2431.2	<0.001	2906.6 ± 2864.2	3899.5 ± 3034.0	<0.001
Admission to ward (no ICU), *N* = 2179	1546	633		1546	357		1546	119	
LOS in ED (h)	14.66 ± 14.66	12.91 ± 12.11	0.004	14.66 ± 14.66	10.52 ± 10.26	<0.001	14.66 ± 14.66	14.89 ± 13.09	0.866
Costs in ED (NT$)	8303.4 ± 7860.9	7092.4 ± 6242.9	<0.001	8303.4 ± 7860.9	6469.9 ± 5682.0	<0.001	8303.4 ± 7860.9	7397.9 ± 7184.8	0.223
Admission to ICU, *N* = 41	33	8		33	5		33	1	
LOS in ED (h)	7.84 ± 10.10	12.09 ± 10.96	0.299	7.84 ± 10.10	6.90 ± 5.39	0.841	7.84 ± 10.10	25.00 ± 0.00	-
Costs in ED (NT$)	30,935.5 ± 37,952.3	30,410.1 ± 57,536.8	0.974	30,935.5 ± 37,952.3	8796.6 ± 4089.2	0.002	30,935.5 ± 37,952.3	3977.0 ± 0.00	-
With CT, *N* = 13,670									
	No POCUS	POCUS	*p*	No POCUS	POCUS within 1 h	*p*	No POCUS	POCUS above 1 h	*p*
OPD, *N* = 6239	3599	2640		3599	1878		3599	416	
LOS in ED (h)	6.48 ± 6.85	6.35 ± 5.62	0.425	6.48 ± 6.85	5.62 ± 4.89	<0.001	6.48 ± 6.85	7.06 ± 5.87	0.060
Costs in ED (NT$)	10,400.3 ± 3476.8	10,587.9 ± 3282.9	0.029	10,400.3 ± 3476.8	10,253.4 ± 2995.3	0.103	10,400.3 ± 3476.8	11,164.4 ± 4037.2	<0.001
Admission to ward (no ICU), *N* = 5551	3778	1773		3778	1185		3778	318	
LOS in ED (h)	16.73 ± 14.48	15.44 ± 13.28	0.001	16.73 ± 14.48	14.11 ± 12.05	<0.001	16.73 ± 14.48	16.77 ± 13.97	0.962
Costs in ED (NT$)	15,233.5 ± 9822.4	15,196.9 ± 9576.5	0.896	15,233.5 ± 9822.4	14,733.8 ± 9160.1	0.107	15,233.5 ± 9822.4	15,881.6 ± 9935.7	0.259
Admission to ICU, *N* = 196	135	61		135	38		135	14	
LOS in ED (h)	14.19 ± 13.67	16.96 ± 16.54	0.219	14.19 ± 13.67	14.16 ± 14.63	0.992	14.19 ± 13.67	14.91 ± 16.01	0.852
Costs in ED (NT$)	34,591.3 ± 35,910.6	33,618.0 ± 26,153.8	0.831	34,591.3 ± 35,910.6	28,427.4 ± 20,144.1	0.173	34,591.3 ± 35,910.6	34,212.7 ± 30,060.9	0.969

Student’s *t*-test, and chi-square tests, as appropriate; PoCUS: point of care ultrasonography; TWD: new Taiwan dollar; CT: computer tomography; OPD: outpatient disposition; ICU: intensive care unit; ED: emergency department; LOS: length of stay.

**Table 3 diagnostics-15-01580-t003:** Quality of care in POCUS only group (unadjusted).

	Admission After Index Visit, *N* = 7967	PoCUS Within 1 h Alone at Index Visit and Admitted After an Unscheduled Return Visit, *N* = 110	*p*-Value
LOS in ED 1st (h)	15.66 ± 14.12	3.25 ± 3.02	<0.001
LOS in ED 2nd (h)	N/A	13.05 ± 11.48	-
Total LOS in ED	15.66 ± 14.12	16.30 ± 11.83	0.571
ED 1st cost (NT$)	13,782.8 ± 11,808.3	3166.6 ± 1560.2	<0.001
ED 2nd cost (NT$)	N/A	10,889.3 ± 8212.6	-
Admission cost (NT$)	100,911 ± 261,456	51,195.6 ± 43,312.5	<0.001
Total cost	114,618 ± 263,743	65,251.5 ± 45,280.1	<0.001
Hospital LOS (day)	8.91 ± 12.45	5.89 ± 5.36	<0.001
ICU	237 (2.97)	2 (1.82)	0.773
ICU LOS (day)	7.67 ± 10.34	2.00 ± 0.00	<0.001
Expired	0(0.00)	0 (0.00)	-
Expired after admission	306 (3.84)	1 (0.91)	0.131

PoCUS: point-of-care ultrasound; N/A: not applicable; TWD: new Taiwan dollar; CT: computer tomography; OPD: outpatient disposition; ICU: intensive care unit; ED: emergency department; LOS: length of stay.

**Table 4 diagnostics-15-01580-t004:** Quality of care in POCUS only group (adjusted).

Outcome Measures, Point Estimate (95% CI)	Admission After Index Visit *N* = 7967	PoCUS Within 1 h Alone at Index Visit and Admitted After an Unscheduled Return Visit, *N* = 110	*p*-Value
LOS in 1st ED (h)	Ref.	−11.59 (−14.20 to −8.98)	<0.001
Total ED LOS (h)	Ref.	1.47 (−1.15 to 4.09)	0.271
1st ED costs (NT$)	Ref.	−9436.1 (−11,542.9 to −7329.3)	<0.001
Total ED costs (NT$)	Ref.	1458.6 (−655.6 to 3572.8)	0.176
Total cost (including admission costs) (NT$)	Ref.	−32,807.8 (−81,456.9 to 15,841.3)	0.186
ICU, OR	Ref.	0.88 (0.21 to 3.68)	0.855
Expired, OR	Ref.	-	-
Expired after admission, OR	Ref.	0.37 (0.05 to 2.72)	0.331
Hospital LOS (day)	Ref.	−1.77 (−4.03 to 0.50)	0.127

Adjusted for age, gender, triage, BMI, and comorbidities. CI: confidence interval; ED: emergency department; TWD: New Taiwan Dollars; LOS: length of stay; h: hour; OR: odds ratio; ICU: intensive care unit.

## Data Availability

The raw data supporting the conclusions of this article will be made available by the author, Shih-Hao Wu, without undue reservation.

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
