# Peer review of "Point-of-Care Ultrasound Within One Hour Associated with ED Flow and Resource Use in Non-Traumatic Abdominal Pain: A Retrospective Observational Study"

_diagnostics, 2025, doi:10.3390/diagnostics15131580_

Round 1
Reviewer 1 Report
Comments and Suggestions for Authors
This is an interesting and important contribution to the literature on point of care ultrasound.
Main critique:
The biggest limitation (which is not acknowledged in the paper) is that the authors do not seem to specify what organ systems were scanned, what diagnoses were identified, whether the exams were single organ or multi-organ, etc. POCUS is not some homogenous entity that should be thought of as a binary concept of POCUS or no POCUS. Every single organ system to which pocus is applied has unique idiosyncrasies associated with its sensitivity, specificity, and overall clinical utility. Reading this study, I come away with no understanding of what organs were scanned, whether the exams were more accurate for some organs than others, whether multi-organ exams were better or worse than single organ exams, etc. The authors need to add significant commentary to explain what information is known about the exams that were performed, what organ systems were evaluated, and if possible, whether any single organ system or combination of organ systems showed a trend toward more benefit than other organ systems individually or combinations of organ systems.
Author Response
Point-by-point responses to reviewers
Dear editors,
16th June 2025
Thank you for the opportunity to revise our manuscript (Submission ID diagnostics-3636295) entitled “Point-Of-Care Ultrasound Within One Hour Associated with ED Flow and Resource Use in Non-Traumatic Abdominal Pain: A Retrospective Observational Study. ” We have fully addressed the comments and submitted a revised manuscript with point-by-point responses as below:
Reviewer 1
This is an interesting and important contribution to the literature on point of care ultrasound.
Main critique:
- The biggest limitation (which is not acknowledged in the paper) is that the authors do not seem to specify what organ systems were scanned, what diagnoses were identified, whether the exams were single organ or multi-organ, etc.
Response:
Thanks for your suggestion. This is an excellent and insightful question. We have revised the limitation as lines 126 to 128, “Fourth, this study did not focus on a specific organ or system, nor did it aim to confirm a diagnosis. Therefore, the specificity and sensitivity of PoCUS for making specific diagnoses were not investigated.” And revised the section “2.2. Setting and population”, as lines 117 to 120, “In this study, PoCUS was not limited to evaluating a specific organ or confirming a diagnosis. Instead, it was used to address the clinical questions that arise during patient care, often requiring a comprehensive assessment across multiple organs and systems based on the patient's presenting symptoms. ”
- POCUS is not some homogenous entity that should be thought of as a binary concept of POCUS or no POCUS.
Response:
Thank you for this very important and insightful comment. We agree completely. PoCUS is not a monolithic intervention but rather a complex diagnostic tool whose effectiveness is influenced by a number of variables. Treating it as a binary concept can obscure these critical details.
The heterogeneity of PoCUS can be attributed to several key factors, including:
Operator Proficiency: The experience and skill level of the physician performing the scan, ranging from novice learners to expert users.
Examination Protocol: The specific clinical question being asked and the standardization of the scanning protocol (e.g., a focused RUQ scan vs. a multi-organ shock protocol).
Equipment: The type and quality of the ultrasound machine used (e.g., cart-based system vs. handheld device).
Workflow Integration: How the findings are interpreted, documented, and acted upon within the specific departmental workflow, including quality assurance (QA) processes.
To address this crucial point, we will make the following revisions to our manuscript:
Operator Proficiency: We have revised the section “2.2. Setting and population”, as lines 104 to 109, “Since 2012, PoCUS has been a core component of ultrasound training in emergency medicine residency programs in Taiwan, with all residents undergoing hands-on as-sessments led by certified instructors from the Taiwan Society of Ultrasound in Medicine, each with over a decade of experience. In clinical settings, PoCUS was performed by residents or attending emergency physicians, all of whom regularly complete refresher courses to maintain and advance their skills. ”
And revised the limitation as lines 123 to 126,“ Furthermore, the effect of different training levels on diagnostic accuracy has not been evaluated. However, PoCUS examinations performed by residents were supervised by attending physicians, which may have reduced the impact of differences in training on the results.”
Examination Protocol: We have revised the section “2.2. Setting and population”, as lines 120 to 122, “All physicians at this tertiary center had received standardized PoCUS training and could apply any protocol or technique they had learned to guide their clinical decisions. ”
Equipment: We have revised the section “2.2. Setting and population”, as lines 110 to 113, “The ED is equipped with one mobile GE Vivid iq, three mobile GE Venue units, three mobile GE LOGIQ™ e units, one fixed GE LOGIQ E9 XDclear 2.0 system, and one fixed GE LOGIQ S7 General Imaging system. All machines undergo routine maintenance every three months, and technical support is available immediately in case of malfunction. ”
Workflow Integration: We have revised the section “2.2. Setting and population”, as lines 113 to 116, “Ultrasound images can be uploaded wirelessly to the Picture Archiving and Communication System (PACS) in real time. Reports can be typed directly into the Hospital Information System (HIS), ensuring efficient documentation and accessibility.”
- Every single organ system to which pocus is applied has unique idiosyncrasies associated with its sensitivity, specificity, and overall clinical utility. Reading this study, I come away with no understanding of what organs were scanned, whether the exams were more accurate for some organs than others, whether multi-organ exams were better or worse than single organ exams, etc.
Response:
Thank you for highlighting this important issue. Although these results and discussions were initially included in this article, they were removed to maintain focus and avoid excessive length following discussion within our team. We categorized the cases by discharge diagnosis from the ED and analyzed the effectiveness and distinct advantages of POCUS across diagnostic groups. This analysis also revealed opportunities to enhance our hospital’s educational offerings. The possible final title is 'Targeted Application of Early Point-of-Care Ultrasound for Non-Traumatic Abdominal Pain Shows Diagnosis-Specific Advantages in the ED'. It has not yet been submitted. We also mentioned it in the conclusion, as lines 147 to 149, “ Future research should validate these results across diverse healthcare settings and examine its effectiveness in various diagnostic groups within the ED.”
- The authors need to add significant commentary to explain what information is known about the exams that were performed, what organ systems were evaluated, and if possible, whether any single organ system or combination of organ systems showed a trend toward more benefit than other organ systems individually or combinations of organ systems.
Response:
Thank you for this important comment. We fully agree with your concern. As epigastric pain can be caused by cardiovascular emergencies, small bowel disease, or diseases of the biliary tract, a systemic approach was crucial. We have therefore revised the section entitled '2.2. Setting and Population', as mentioned above on lines 117 to 120, “In this study, PoCUS was not limited to evaluating a specific organ or confirming a diagnosis. Instead, it was used to address the clinical questions that arise during patient care, often requiring a comprehensive assessment across multiple organs and systems based on the patient's presenting symptoms.”
We believe that the updated manuscript has substantially improved with the reviewers’ comments and hope it will now be suitable for publication. Many thanks for your kind consideration.
Best wishes,
Authors
Reviewer 2 Report
Comments and Suggestions for Authors
It is seen that the authors retrospectively investigated the relationship between emergency room flow and resource usage in non-traumatic abdominal pain and ultrasonography performed at the point of care within one hour. I want to state that it is well written structurally and visually. However, I would also like to state that there are some deficiencies.
1. Abstract: Lines 13-17 were unsuitable for the abstract. When it comes to previous studies, it is necessary to add literature. However, an introduction suitable for your study is necessary. Lines 18-19 can be included in the method section rather than the abstract. It would be appropriate to start the results with the age and gender of the cases. In addition, instead of p<0.005, the real p value should be added.
2. The iThenticate report is 27% high; it would be appropriate to reduce it to below 20%.
3. Introduction: What are the causes of non-traumatic abdominal pain? Is it only gallbladder and kidney stones? It should be explained. PoCUS indications should be explained in detail. What are its advantages and disadvantages compared to CT?
4. Method: Patients' acceptance and rejection criteria should be written clearly and separately. In addition, the period when the study was conducted was one of the most severe periods of COVID-19 infection. How did you distinguish these patients? Did the cases you examined with FoCUS have COVID-19 complications? There is a very large patient population. Which power analysis did you use? What was the margin of error and acceptability rate? Were the data used in the study emergency room data or radiology data? Is there an emergency medicine specialist in this large hospital? Who made the decision? Explanation.
5. Findings: It seems imaginary that all the data in Table 1 are statistically significant and far from reality.
Table 2 is both very complicated, and it should be stated which statistical analysis the p-value used corresponds to.
Author Response
Point-by-point responses to reviewers
Dear editors,
16th June 2025
Thank you for the opportunity to revise our manuscript (Submission ID diagnostics-3636295) entitled “Point-Of-Care Ultrasound Within One Hour Associated with ED Flow and Resource Use in Non-Traumatic Abdominal Pain: A Retrospective Observational Study. ” We have fully addressed the comments and submitted a revised manuscript with point-by-point responses as below:
Reviewer 2
It is seen that the authors retrospectively investigated the relationship between emergency room flow and resource usage in non-traumatic abdominal pain and ultrasonography performed at the point of care within one hour. I want to state that it is well written structurally and visually. However, I would also like to state that there are some deficiencies.
1.Abstract:
- Lines 13-17 were unsuitable for the abstract. When it comes to previous studies, it is necessary to add literature. However, an introduction suitable for your study is necessary.
Response:
Thanks for your suggestion. We have revised the abstract as lines 13 to 15 “Although the value of point-of-care ultrasonography (PoCUS) is well-established for specific diseases and in the hands of trained users, its broader impact on overall ED efficiency is not yet fully known.”
- Lines 18-19 can be included in the method section rather than the abstract.
Response:
Thanks for your suggestion. We have revised the abstract as lines 18 to 19 “This retrospective cohort study included 44,863 adult patients (≥18 years) presenting with non-traumatic abdominal pain from January 2021 to December 2023.”
- It would be appropriate to start the results with the age and gender of the cases.
Response:
Thanks for your suggestion. We have revised the abstract as lines 23 to 24 “The mean age of the subjects was 44.4 ± 17.9 years, and 61.2% were female. Point-of-care ultrasound (PoCUS) was performed in 39.7% of cases, with 69.6% of these conducted within one hour. ”
- In addition, instead of p<0.005, the real p value should be added.
Response:
Thank you for your reminder. We have revised the abstract to include all p-values, as shown in lines 25 to 30.“The PoCUS group had a significantly shorter ED LOS compared to the no-PoCUS group among patients admitted to general wards (p < 0.001), but not in outpatient dispositions (p = 0.282) or ICU admissions (p = 0.081). Subgroup analysis of patients receiving PoCUS within 1 hour showed a significantly shorter LOS for both outpatient dispositions (p < 0.001) and general ward admissions (p < 0.001), with no effect on ICU admissions (p = 0.869). ”
- The iThenticate report is 27% high; it would be appropriate to reduce it to below 20%.
Response:
Thank you for your reminder. We have revised some paragraphs, particularly the method, to reduce it to below 20%.
- Introduction:
- What are the causes of non-traumatic abdominal pain? Is it only gallbladder and kidney stones? It should be explained.
Response:
Thank you for your reminder. We have revised the introduction, as shown in lines 45 to 47.“ It encompasses a wide range of potential causes, from benign, self-limiting conditions to life-threatening emergencies such as bowel obstruction, mesenteric ischaemia, or ruptured abdominal aortic aneurysm [2]. ”
- PoCUS indications should be explained in detail.
Response:
Thank you for your reminder. We have added PoCUS indications in detail in the Introduction, as shown in lines 58 to 59.”PoCUS is a bedside diagnostic tool that allows clinicians to acquire, interpret, and im-mediately integrate ultrasound imaging into patient care [11].” and in the section “2.2. Setting and population”, as shown in lines 117 to 122.“In this study, PoCUS was not limited to evaluating a specific organ or confirming a diagnosis. Instead, it was used to address the clinical questions that arise during patient care, often requiring a comprehensive assessment across multiple organs and systems based on the patient's presenting symptoms. All physicians at this tertiary center receive standardized PoCUS training and are free to apply any protocol or technique they have learned to guide their clinical decisions. ”
- What are its advantages and disadvantages compared to CT?
Response:
Thank you for your reminder. We have revised the Introduction, as shown in lines 69 to 75.“The benefits of PoCUS must be weighed against other imaging modalities. Computed tomography (CT) provides superior diagnostic accuracy [19] and has been shown to enhance surgical decision-making in cases of acute abdominal pain [20]. However, CT carries notable drawbacks, including higher costs and significant radiation exposure [18]. In contrast, PoCUS is more accessible and avoids radiation but is highly operator-dependent [21]. Its diagnostic performance var-ies with patient factors, operator expertise, and equipment quality [22]. ”
- Method:
- Patients' acceptance and rejection criteria should be written clearly and separately.
Response:
Thank you for your suggestion. We have revised the section “2.2. Setting and population”, as shown in lines 97 to 102.“The patients were included if they (i) had an ED visit with no prior ED visit or hospitalization in the preceding three days; (ii) had non-traumatic abdominal pain as the chief complaint at triage; (iii) were adults aged 18 years or older. Patients were excluded if they were (i) younger than 18 years; (ii) had traumatic abdominal pain; (iii) were transferred from other EDs; or (iv) were discharged against medical advice from other clinics or hospitals. The study included 44,863 patients.”
- In addition, the period when the study was conducted was one of the most severe periods of COVID-19 infection. How did you distinguish these patients?
Response:
Thank you for this very important and insightful comment. The COVID-19 epidemic broke out during the period when we collected data. During the COVID-19 epidemic, ED visits declined sharply, and management processes were impacted by universal prevention protocols, enhanced disinfection routines, and other pandemic-related measures. In our country, patients presenting with fever or respiratory symptoms are first screened outdoors with a PCR test for SARS-CoV-2 to minimize exposure risks. Only those who test negative are admitted to the ED’s safe zone; those who test positive are directed to a separate area managed by emergency physicians.
Patients presenting solely with abdominal pain and no respiratory symptoms are admitted directly to the safe zone for evaluation. Our study focuses on this group—patients whose primary complaint is abdominal pain. Those who also exhibit fever or respiratory symptoms are tested for SARS-CoV-2 prior to entering the ED. Notably, among all study participants, 212 tested positive for the virus.
In line with your reminder, we have revised the limitation as lines 129 to 133, “Fifth, part of this study was conducted during the peak of the COVID-19 pandemic, when ED visits declined significantly and infection control measures, such as universal precautions and enhanced disinfection, disrupted normal workflows. As a result, the timing of PoCUS may have deviated slightly from standard practice. However, the clinical approach to non-traumatic abdominal pain remained unchanged.”
- Did the cases you examined with PoCUS have COVID-19 complications?
Thank you for highlighting this important issue. We agree with your concern about whether patients examined with PoCUS have complications from the virus. Of the 44,863 patients included in the study, 17,819 underwent PoCUS examination, of whom only 34 had a confirmed case of SARS-CoV-2 infection. There were no deaths or ICU admissions from the ED. Only seven patients were hospitalised. Nevertheless, there were no deaths, and all patients were discharged from the ward with an outpatient disposition. There was only one revisit, who was not admitted after the return visit.
However, of the 44,863 patients, 212 tested positive for SARS-CoV-2 infection.
None of these 212 patients had mortality or ICU admission from the ED. Only 20 patients were admitted, six of whom had PoCUS. Of these 20 patients, two were discharged against medical advice in critical condition: one was 97 years old, and the other was 94 years old. Three patients died: one was 96 years old, one was 90 years old, and one was a 56-year-old female with a history of left breast infiltrating ductal carcinoma with bone, lung and liver metastasis. All five of these patients did not receive PoCUS. There was only one admission for 23 days after the return visit. He was a liver transplant case and was discharged as an outpatient from the ward without POCUS at the index visit.
In line with your suggestion, we have revised the limitation as lines 133 to 136, “Notably, 17,819 patients underwent PoCUS examination, of whom only 34 had a confirmed SARS-CoV-2 infection. No complications were observed in SARS-CoV-2-positive patients who underwent PoCUS.”
- There is a very large patient population. Which power analysis did you use?
Response:
Thank you for your reminder. We have mentioned it, as shown in lines 159 to 161. “We used multivariable logistic and linear regression models to adjust for differences in patient mix. Potential confounding factors included age, gender, triage, BMI, and comorbidities. ”
- What was the margin of error and acceptability rate?
Response:
Thank you for your reminder. We have mentioned it, as shown in lines 161 to 163. “All odds ratios (OR) and beta-coefficients are presented with 95% CIs. We performed all analyses using SAS software (version 9.4; SAS Institute Inc., Cary, NC, USA). All P values are two-sided, with P <0.05 considered statistically significant.”
- Were the data used in the study emergency room data or radiology data?
Response:
Thank you for your reminder. The description about the data source was on lines 88 to 90, “This retrospective cohort study used data from the electronic medical record (EMR) of a tertiary medical center in Taiwan with more than 160,000 ED visits annually, between 1 January 2021 and 31 December 2023.” And lines on 103 to 108 “Since 2012, PoCUS has been a core component of ultrasound training in emergency medicine residency programs in Taiwan, with all residents undergoing hands-on assessments led by certified instructors from the Taiwan Society of Ultrasound in Medicine, each with over a decade of experience. In clinical settings, PoCUS was performed by residents or attending emergency physicians, all of whom regularly complete refresher courses to maintain and advance their skills.”
- Is there an emergency medicine specialist in this large hospital?
Response:
Thank you for your reminder. As mentioned above, as shown in lines 108 to 109. “PoCUS was performed by residents or attending emergency physicians.” We have revised the section “2.2. Setting and population”, as lines 103 to 104,“This tertiary center employed 42 emergency medicine specialists and 19 residents, with all attending physicians certified in emergency medicine. ”
- Who made the decision? Explanation.
Response:
Thank you for your reminder. As mentioned above, as shown in lines 108 to 109. “ PoCUS was performed by residents or attending emergency physicians.” And as mentioned above, as shown in lines 58 to 59. ”PoCUS is a bedside diagnostic tool that allows clinicians to acquire, interpret, and immediately integrate ultrasound imaging into patient care [11].” And as shown in lines 124 to 126.“PoCUS examinations performed by residents were supervised by attending physicians, which may have reduced the impact of differences in training on the results.”Therefore, residents or attending emergency physicians made the decision and explanation.
5. Findings:
- It seems imaginary that all the data in Table 1 are statistically significant and far from reality.
Response:
Thank you for your reminder. We have deleted the P value of vital signs, including heart rate, respiratory rate, body temperature, and blood pressure. As shown on lines 176 to 177, ”Compared with the No-PoCUS group, patients received PoCUS were significantly younger, predominantly female, and more likely to be triaged at level 3. ” These results demonstrate that PoCUS was performed more frequently on younger patients and those with less severe disease, and that fewer medical resources were used. They also show that, in practice, emergency physicians use PoCUS for less severe cases. This may be due to the operator having more confidence.
- Table 2 is both very complicated, and it should be stated which statistical analysis the p-value used corresponds to.
Response:
Thank you for your reminder. We have added the sentence below the table, “Student’s t-test, and chi-square tests, as appropriate.”
We believe that the updated manuscript has substantially improved with the reviewers’ comments and hope it will now be suitable for publication. Many thanks for your kind consideration.
Best wishes,
Authors
Round 2
Reviewer 1 Report
Comments and Suggestions for Authors
The authors have addressed my critiques.
Reviewer 2 Report
Comments and Suggestions for Authors
It is seen that the author made the requested changes. I think the review will attract more attention after the changes. The review can be accepted as it is. I thank the editor and authors for their contributions.